# Advanced Dual-Satellite Method for Detection of Low Stratus and Fog near Japan at Dawn from FY-4A and Himawari-8

**Jung-Hyun Yang [1], Jung-Moon Yoo [2],\* and Yong-Sang Choi [1]**

[1] Department of Atmospheric Science and Engineering, Ewha Womans University, Seoul 120-750, Korea; junghyunyang@ewhain.net (J.-H.Y.); ysc@ewha.ac.kr (Y.-S.C.)

[2] Department of Science Education, Ewha Womans University, Seoul 120-750, Korea

\* Correspondence: yjm@ewha.ac.kr; Tel.: +82-02-3277-4671

**Abstract:** The detection of low stratus and fog (LSF) at dawn remains limited because of their optical features and weak solar radiation. LSF could be better identified by simultaneous observations of two geostationary satellites from different viewing angles. The present study developed an advanced dual-satellite method (DSM) using FY-4A and Himawari-8 for LSF detection at dawn in terms of probability indices. Optimal thresholds for identifying the LSF from the spectral tests in DSM were determined by the comparison with ground observations of fog and clear sky in/around Japan between April to November of 2018. Then the validation of these thresholds was carried out for the same months of 2019. The DSM essentially used two traditional single-satellite tests for daytime such as the 0.65-μm reflectance ($R_{0.65}$), and the brightness temperature difference between 3.7 μm and 11 μm ($BTD_{3.7-11}$); in addition to four more tests such as Himawari-8 $R_{0.65}$ and $BTD_{13.5-8.5}$, the dual-satellite stereoscopic difference in $BTD_{3.7-11}$ ($\Delta BTD_{3.7-11}$), and that in the Normalized Difference Snow Index ($\Delta NDSI$). The four were found to show very high skill scores (POD: $0.82 \pm 0.04$; FAR, $0.10 \pm 0.04$). The radiative transfer simulation supported optical characteristics of LSF in observations. The LSF probability indices (average POD: 0.83, FAR: 0.10) were constructed by a statistical combination of the four to derive the five-class probability values of LSF occurrence in a grid. The indices provided more details and useful results in LSF spatial distribution, compared to the single satellite observations (i.e., $R_{0.65}$ and/or $BTD_{3.7-11}$) of either LSF or no LSF. The present DSM could apply for remote sensing of environmental phenomena if the stereoscopic viewing angle between two satellites is appropriate.

**Keywords:** fog; low stratus; dual satellite method; Himawari-8; Fengyun-4A

---

## 1. Introduction

Reliable monitoring of low stratus and fog (LSF) is of vital importance because restricted visibility can cause hazards in transportation and navigation including LSF-related accidents (e.g., [1,2]). Thick LSF at dawn is a major hazard during the morning rush hours. The persistence and frequency of LSF on land can be associated with various factors: air pollution and quality [3–5] and climate variabilities such as Arctic Oscillation [6] and El Niño/Southern Oscillation [7,8].

According to Yoo et al. [9], 71–76% of fog phenomena in surface synoptic observations (SYNOP) tend to accompany the low stratus over the Korean Peninsula at summertime dawn. This indicates that the better the LSF detection, the higher the possibility for fog detection. Low stratus and fog are often decoupled from the upper part of fog at night. Thus, low stratus and fog are examined in terms of LSF without individual distinctions in this study. LSF is also an appropriate term in satellite remote sensing since the distinction between low stratus and fog is fundamentally difficult without additional ground information such as surface elevation and cloud base height, etc. [2,3,10–12]. The only difference between the two weather phenomena is whether their bases are in contact with the ground [13].

Since LSF has various scales in space and time [6,14], its detection from both ground observations and numerical forecast models has intrinsic weaknesses. The observations cannot reflect spatial distributions of LSFs because of a limited number of stations [15]. The models cannot simulate LSFs as fast as needed due to their long spin-up time [16], and with inaccuracy due to coarse spatiotemporal resolutions (e.g., [2,17]) and limited parameterizations of microphysical and turbulent mixing processes (e.g., [1,18]). The most important measure for continuous monitoring of LSF over a wide area is geostationary satellites (GEOs). However, it is also regarded that the sensing accuracy in LSF at dawn is in need of improvement.

Traditional detection methods for LSF from a single satellite utilize thresholds that come from either various satellite channels or their spectral differences [19]. During the nighttime, Eyre et al. [20] developed a fog detection method by the difference in brightness temperature at 3.7 μm and 11 μm ($BTD_{3.7-11}$) from the Advanced Very High-Resolution Radiometer (AVHRR), using the principle that 3.7 μm has a lower emissivity for water droplets than 11 μm. The $BTD_{3.7-11}$ threshold test for LSF from GEOs has been continuously used [21–27].

During the daytime, $BTD_{3.7-11}$ is less useful because the brightness temperature at ~3.7 μm ($BT_{3.7}$) includes both the solar and earth radiation. Thus, the visible 0.68 μm reflectance ($R_{0.68}$) test has been attempted instead (e.g., [28]), based on that a water droplet has a higher reflectivity than the clear sky. Kim et al. [29] have carried out the daytime marine fog detection based on the decision tree technique from the Geostationary Ocean Color Imager (GOCI) and Himawari-8 Advanced Himawari Imager (AHI); although the hit rate (HR) is 0.66–1.00, there is room for improvement due to the high false alarm ratio (FAR) and the limited amount of data at three Korean islands (FAR = 0.31–0.33). The visible technique has obvious drawbacks for distinguishing LSF from mid/high-level clouds. Thereby, researchers have tried to overcome this hindrance using both visible and infrared thresholds [14,28,30].

Despite many studies on LSF remote sensing [31,32], the detection accuracy remains low, particularly at dawn [5,33]. The $BTD_{3.7-11}$ value at dawn is nearly zero due to rapidly changing optical features with different signs (i.e., negative to positive) [34]. Like snow, the $R_{0.67}$ value at dawn also increases in inaccuracy depending on SZA or surface conditions [33,35]. To address this issue, Yoo et al. [9] and Yang et al. [36] proposed the dual-satellite method (DSM) that utilizes the observed stereoscopic differences (Δ) in $R_{0.67}$ and $BTD_{3.7-11}$, between two GEOs (i.e., COMS and FY-2D), defining $\Delta R_{0.67}$ and $\Delta BTD_{3.7-11}$ in this study. Both studies applied DSM to LSF detection at dawn near the Korean Peninsula. While the $\Delta R_{0.67}$ test for LSF detection was useful in the summertime [9], the advantage of the $\Delta BTD_{3.7-11}$ test for the detection was emphasized in the springtime [36].

An example of the basic principle of DSM is that the detection of an object is better by two eyes (or satellites) than a single eye [37]. Since DSM for LSF detection is based on the bidirectional reflectance function (BRDF) of the two GEOs, it is most useful in summertime dawn when the stereoscopic differences are maximized under the Viewing Zenith Angle difference (ΔVZA) of ~46.5° across the Korean Peninsula [9]. However, the accuracy for LSF detection highly depended on ΔVZA, season, and satellite-observed channels. Therefore, new tests are required for the LSF detection improvement, utilizing the additional channels and high spatiotemporal resolution of the recently advanced imagers of GEOs. The purpose of this study was to improve the detection accuracy and spatial information of LSF at dawn near Japan in terms of probability index (PI) by devising additional tests from the advanced satellites, Feng-Yun-4A (FY-4A; [38,39]) and Himawari-8 [40] to the previous tests [9,36]. Compared to previous GEOs, these two GEOs have many channels in high spatiotemporal resolution, and they can provide additional independent information. The two GEOs have ΔVZA of ~40.4° in the region of interest near the nadir of Himawari-8 (Figure 1). The selection of the region is reasonable to enhance the signal-to-noise ratio in DSM. Fog in Japan occurs more frequently during the warm season than the cold season, with high regional variability [41].

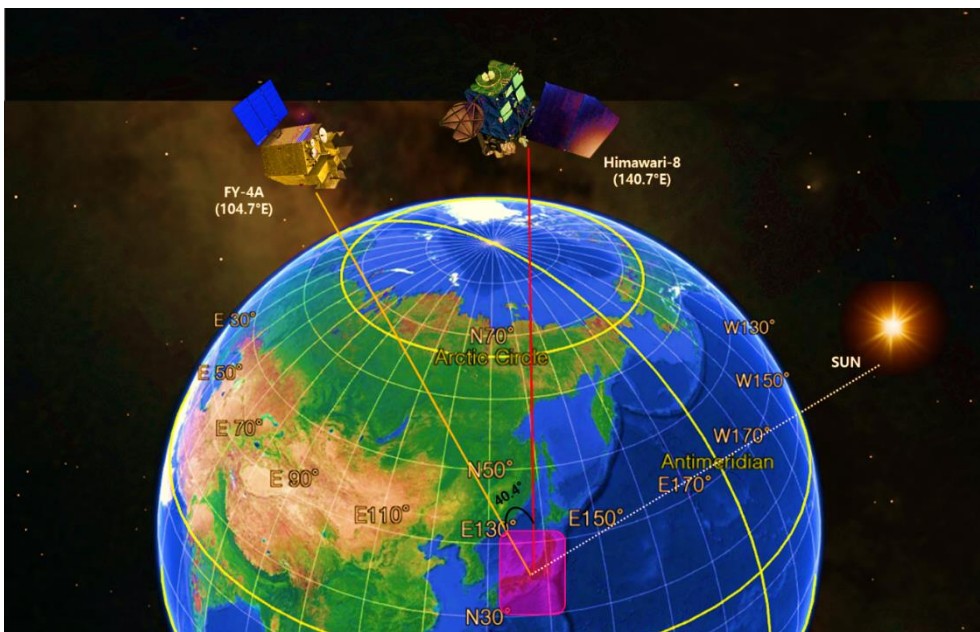

**Figure 1.** Viewing zenith angles (VZAs) of two geostationary satellites (Himawari-8 and FY-4A) available for near-simultaneous observations of the low stratus and fog (LSF) at dawn near Japan. The VZA difference between the satellites is 40.4° at the Himawari-8 nadir point.

The Radiative Transfer Model (RTM) simulations of this study which were arranged and stored in the Look-Up Table (LUT) in advance, were comprehensively used for an approximate diagnostic check of the LSF optical properties. In Section 2, the data and method were described. Here spectral properties of newly developed tests were explained. The results were shown in terms of the LSF and clear-sky classification using optimal thresholds with radiative transfer simulations in Section 3. The case study for validation was shown with the PI formulation. Discussion and conclusions were presented in Sections 4 and 5, respectively.

## 2. Materials and Methods

To derive and validate the threshold values for LSF detection at dawn near Japan, the following three datasets were used: geostationary satellite observations from Himawari-8 and FY-4A, ground observations from Meteorological Aerodrome Report (METAR) for the weather phenomena of fog and clear sky, and the LUT from RTM. For this study, "dawn" was defined as within 2 h after sunrise (67° < SZA < 86°). The detailed information for the datasets is described below. For SZA, the average of LSF cases in 2018 was ~79° and that of clear-sky conditions was ~78°. In 2019, both SZA values were ~80°. Also, the meanings of acronyms used in this study are noted in Table A1.

### 2.1. Satellite Observations

The geostationary satellites of Himawari-8 and FY-4A are located at different longitudes over the equator, and they show a VZA difference of 40.4° near the nadir of Himawari-8 (Figure 1). The satellite information and application in this study with respect to the channel are described in Table 1. The primary payloads of FY–4A and Himawari-8 are the Advanced Geostationary Radiation Imager (AGRI; [38,39]) and AHI; [40], respectively. The instruments provided significant advancements in terms of the number of bands, spatial resolution, and temporal frequency. Their solar and thermal channels are separately presented in the table. There were two visible (VIS) channels of solar radiation centered at ~0.65 μm and ~1.6 μm. In the 0.65 μm channel, clouds appeared bright due to their high reflectance, while the signals from the land/sea surface were generally weak. The 1.6 μm channel was in the infrared (IR) region and allowed distinction between snow/ice-covered

land and signals from other sources. This channel was sensitive to the cloud phase, and water clouds reflected more energy in this spectral range than the ice clouds.

**Table 1.** Information from the dual geostationary satellites of Himawari-8 and FY-4A near Japan, available for near-simultaneous observations of low stratus and fog (LSF) at dawn during 2018–2019. The meaning of acronyms is explained in Table A1.

| Channel (Abbreviation) | Application of this Study | Satellite (Nation, Lon at nadir) | | | | | |
| --- | --- | --- | --- | --- | --- | --- | --- |
| | | Himawari-8 (Japan, 140.7 °E) | | | FY-4A (China, 104.7 °E) | | |
| | | Wavelength (μm) | Central Wavelength (μm) | Resolution at Nadir (km) | Wavelength (μm) | Central Wavelength (μm) | Resolution at Nadir (km) |
| VIS ($R_{0.65}$) | Fog, cloud | 0.63–0.66 | 0.64 | 0.5 | 0.55–0.75 | 0.65 | 0.5–1 |
| NIR ($R_{1.6}$) | Cloud, snow | 1.60–1.62 | 1.61 | 2 | 1.58–1.64 | 1.61 | 2 |
| SWIR ($BT_{3.7}$) | Fog, cloud | 3.74–3.96 | 3.89 | 2 | 3.50–4.00 | 3.75 | 2 |
| MIR ($BT_{8.5}$) | Cloud-top phase, SST, LST | 8.44–8.76 | 8.59 | 2 | 8.00–9.00 | 8.50 | 4 |
| LWIR1 ($BT_{11}$) | SST, LST, cloud-top temperature | 11.1–11.3 | 11.24 | 2 | 10.3–11.3 | 10.80 | 4 |
| LWIR3 ($BT_{13.5}$) | Cloud, air temperature | 13.2–13.4 | 13.28 | 2 | 13.2–13.8 | 13.50 | 4 |

Of the six channels used, the four non-solar channels were classified as thermal parts. The 3.7 μm lies in the spectrum of two outgoing energy sources (e.g., solar reflectance and thermally emitted radiation). This channel is not only sensitive to the cloud phase but also useful for the detection of water clouds at night. In the region around 8.5 μm, there is a little water vapor absorption, but it is still in the window region. The IR window channel around 11 μm is affected primarily by radiation from the Earth's surface or cloud tops. The 13.3–13.5 μm band is used as cloud masks and cloud-top height.

Level 1B data as digital count values were used to convert the VIS and near-infrared (NIR) data to visible reflectance and the others (mid-IR, longwave-IR1, longwave-IR2, longwave-IR3) to brightness temperature. In order to observe the two satellite observations on the same-sized grid, collocation was performed by interpolating the Himawari-8 observations to FY-4A along with the ground observations shown in Table 2 and Figure 2. The analysis of this study is not so sensitive to the grid-size of either 4 km by 4 km or the original pixel (2 km by 2 km), but the data smoothing effect tends to increase in the former grid. To compare the dual-satellite observations by pixel, the spatial resolution at the channels noted above was set to 4 km by 4 km. Thus, the Himawari-8 data in a two-dimensional array of longitude and latitude near Japan (122.5-144E, 23.5-44N) were interpolated into the FY-4A array. This method keeps the FY-4A accuracy without degradation. The observations for seven variables ($BTD_{13.5-8.5}$, $R_{0.65}$, $BTD_{3.7-11}$, NDSI, $\Delta R_{0.65}$, $\Delta$NDSI, and $\Delta BTD_{3.7-11}$) were analyzed for LSF detection. Additionally, the difference value between Land Surface Temperature (LST) and 11 μm brightness temperature ($BT_{11}$) was used to analyze cloud-top height as an acronym of LST-$BT_{11}$. In this study, $BTD_{13.5-8.5}$ was the brightness temperature difference between 13.5 μm and 8.5 μm. Based on the main weighting function peak in the Satellite Infrared Spectrometer (SIRS-B) channel at 750 cm$^{-1}$ [42], the peak of AHI at 13.28 μm (753 cm$^{-1}$) was located near the 850–900 hPa level. Meanwhile, the 8.5 μm AHI surface channel was used for the analysis of the cloud phase. Reflectance values ($R_{0.6}$ and $R_{1.6}$) were normalized by dividing by cos(SZA) and set to unity when the values were greater than 1 due to normalization (e.g., large SZA $\geq$ ~80°). The difference in values between the two collocated satellites is defined with Equations (1)–(3) as follows:

$$\Delta R_{0.65} = R_{0.65}^{FY\text{-}4A} - R_{0.65}^{Himawari\,text\,8} \tag{1}$$

$$\Delta BTD_{3.7\text{-}11} = BTD_{3.7-11}^{FY\text{-}4A} - BTD_{3.7-11}^{Himawari\text{-}8} \tag{2}$$

$$\Delta NDSI = NDSI^{FY\text{-}4A} - NDSI^{Himawari\text{-}8} \tag{3}$$

**Table 2.** Thirty-one meteorological stations of the Automatic Synoptic Observing System (ASOS) near Japan used for the LSF analysis in this study.

| Station Number | Station in Japan | Lat (°N) | Lon (°E) | Height (m) | Station Number | Station in Japan | Lat (°N) | Lon (°E) | Height (m) |
|---|---|---|---|---|---|---|---|---|---|
| 1 | AKENO (JASDF) | 34.53 | 136.67 | 9 | 17 | KUMAMOTO (CIV/JAS) | 32.84 | 130.86 | 196 |
| 2 | AKITA AIRPORT | 39.62 | 140.22 | 96 | 18 | KUSHIRO AIRPORT | 43.04 | 144.19 | 98 |
| 3 | AOMORI AIRPORT | 40.73 | 140.69 | 201 | 19 | MATSUYAMA AIRPORT | 33.83 | 132.70 | 7 |
| 4 | ASAHIKAWA AIRPORT | 43.67 | 142.45 | 211 | 20 | MIYAZAKI AIRPORT | 31.88 | 131.45 | 9 |
| 5 | CHITOSE | 42.77 | 141.69 | 30 | 21 | NEW HIROSHIMA | 34.44 | 132.92 | 6 |
| 6 | FUKUI AIRPORT | 36.14 | 136.22 | 8 | 22 | NEW TOKYO INTL AIRPORT | 35.77 | 140.39 | 44 |
| 7 | FUKUOKA/ ITAZUKE | 33.58 | 130.45 | 12 | 23 | OKAYAMA AIRPORT | 34.76 | 133.86 | 242 |
| 8 | FUKUSHIMA ARPT | 37.23 | 140.43 | 375 | 24 | OMINATO (JASDF) | 41.23 | 141.13 | 10 |
| 9 | HACHINOHE (JMSDF) | 40.55 | 141.47 | 49 | 25 | SAGA AIRPORT | 33.15 | 130.30 | 5 |
| 10 | HAKODATE AIRPORT | 41.77 | 140.82 | 36 | 26 | SENDAI AIRPORT | 38.14 | 140.92 | 5 |
| 11 | HYAKURI (JASDF) | 36.18 | 140.41 | 35 | 27 | SHIMOFUSA (JMSDF | 35.80 | 140.01 | 33 |
| 12 | IZUMO AIRPORT | 35.41 | 132.89 | 5 | 28 | TAKAMATSU AIRPORT | 34.22 | 134.02 | 188 |
| 13 | KAGOSHIMA AIRPORT | 31.80 | 130.72 | 275 | 29 | TOKUSHIMA (JMSDF) | 34.13 | 134.61 | 11 |
| 15 | KANSAI INTL | 34.43 | 135.23 | 8 | 30 | Yonaguni | 24.47 | 122.98 | 19 |
| 15 | KISARAZU (JGSDF) | 35.40 | 139.91 | 6 | | **Island station in South Korea** | | | |
| 16 | KOCHI AIRPORT | 33.55 | 133.67 | 10 | 31 | Ulleung | 37.48 | 130.90 | 223 |

NDSI [43] was calculated from the visible reflectance values of 0.65 μm and 1.6 μm as shown in Equation (4).

$$NDSI = \frac{R_{0.65} - R_{1.6}}{R_{0.65} + R_{1.6}} \tag{4}$$

Four variables ($BTD_{13.5\text{-}8.5}$, $R_{0.65}$, $BTD_{3.7\text{-}11}$, and NDSI) were obtained from a single satellite's (i.e., Himawari-8 only) observations, while three variables ($\Delta R_{0.65}$, $\Delta NDSI$, and $\Delta BTD_{3.7\text{-}11}$) were derived from both satellites. Also, LST data were given from ground observations.

The NDSI index has been widely used to distinguish between snow and clouds since snow has a lower reflectance at 1.6 μm [43]. The studies [14,44] presented the NDSI values for various surface conditions (e.g., water clouds, ice clouds, and vegetation,) to eliminate snow pixels, which have relatively large $R_{0.65}$ values. Wu and Li [45] determined both the NDSI and $R_{0.67}$ thresholds for daytime sea fog detection using MODIS data. In their study, the NDSI was used for LSF detection over inland China to divide vegetation and clouds during the warm season, because vegetation has larger $R_{1.6}$ and smaller $R_{0.65}$ than clouds. Ryu and Hong [46] proposed a regression-based method for sea fog detection from a combination of NDSI and reflectance in the AHI green band (0.51 μm).

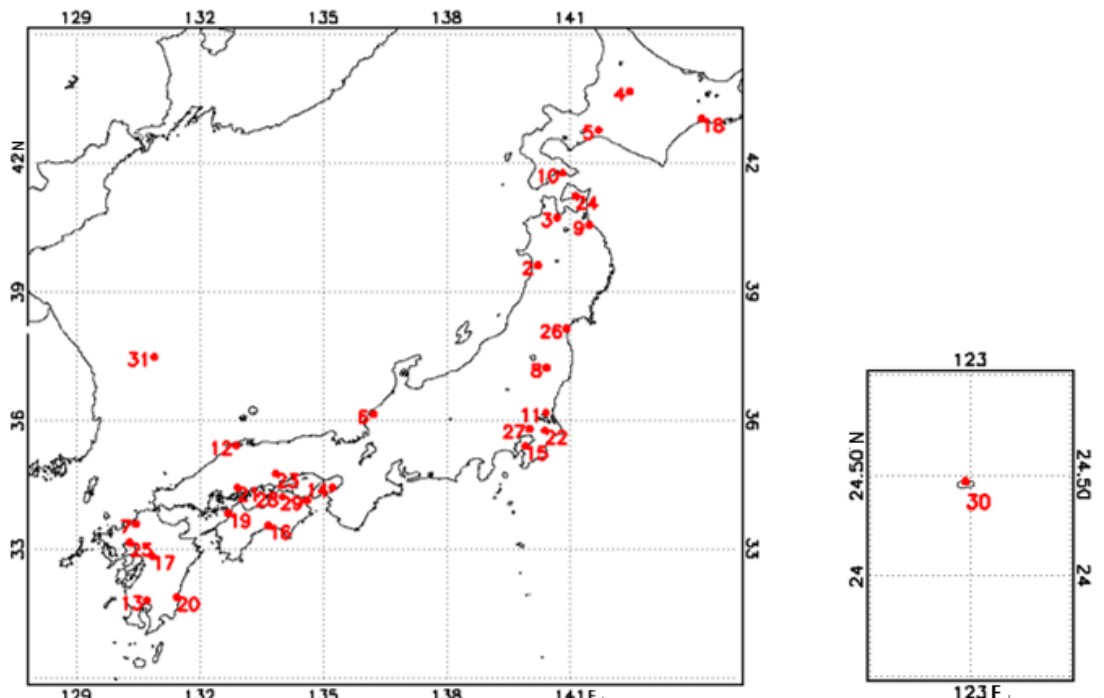

**Figure 2.** Locations of 31 meteorological stations near Japan used for LSF verification during 2018–2019. The stations are located in Japan except for the 31st Ulleung station in South Korea.

The $BTD_{13.5-8.5}$ value which is the difference in brightness temperature between 13.5 and 8.5 μm was used to detect higher ice clouds [47,48], because the 13.5 μm channel belongs to a $CO_2$ absorption band that affects atmospheric transmission for the calculation of cloud altitudes. Thus, this value is sensitive to atmospheric altitude. This result indicates that the $BTD_{13.5-8.5}$ value can be utilized as an indicator to classify LSF among the weather phenomena of clear sky, LSF, and high clouds.

### 2.2. Ground Observations

In this study, ground observation data from METAR were regarded as the ground truth for LSF and clear-sky cases from March to November in 2018 and 2019. The METAR is a weather status report designed for aviation based on information from an Automatic Weather Station (AWS), such as visibility, cloud amount, cloud base height present weather, ambient temperature, and relative humidity at hourly intervals. Figure 2 shows the locations of METAR stations in Japan including the "Ulleung" island station in South Korea. The information for a total of 31 stations over the study region is also described in Table 2. The LSF occurrences were selected only in cases where the current weather conditions were reported as "FG" (i.e., fog). In METAR data, fog appears in various forms depending on accompanying weather phenomena and full or partial spatial covering. Only cases of "FG" that covered the entire observation area without precipitation were selected in this study. The clear-sky cases as a contrast group of LSF cases were adopted under conditions with the abbreviation "CAVOK" or cloud amount with "FEW" (1 to 2 oktas). CAVOK stands for Ceiling (or clouds) And Visibility OK, indicating no clouds below 1500 m or the highest minimum sector altitude and no cumulonimbus or towering cumulus at any level, with a visibility of 10 km or more and no significant weather change [49]. The total observation numbers for LSF and clear-sky occurrences used in this study were 347 and 229 during 2018 and 2019, respectively.

Weather maps (i.e., SYNOP chart) provided by the Korea Meteorological Administration (KMA) were used to verify the presence of LSF and upper cloud information detected by satellites and to obtain extensive information on air pressure patterns and wet areas. However, since the weather maps were produced at three-hour intervals, a direct

comparative analysis was not possible due to the ordinary rapid generation and dissipation of fog. Therefore, weather map analysis was performed only when the map time coincided with the ground or satellite observation time. If there was a time difference between fog occurrence time and weather map, the $BT_{11}$ distribution, which is approximately related to cloud top temperature, was used instead.

### 2.3. Radiative Transfer Model

Optical characteristics of LSF can be estimated using the RTM simulation with satellite observations (e.g., [50]). In this study, the Santo Barbara DISORT (SBDART) of a plane-parallel radiative transfer model was used [51] and the LUT was calculated from RTM in advance for estimation. The cloud model in the SBDART yielded a radiance value relating to cloud optical properties (e.g., Fog Optical Thickness, FOT; Cloud Optical Thickness, COT; Effective Radius, ER) based on the Mie theory. This allowed us to understand satellite-observed spectral values in the LSF environment and their optical difference between the two satellites. In addition, the simulation provided insight into the effects of higher clouds over the LSF layer.

The Spectral Response Function (SRF) difference between the two satellites was less significant than that of the VZA [9]. Therefore, the values at the central wavelength of each channel (https://www.data.jma.go.jp accessed on 3 June 2019 for Himawari-8; http://satellite.nsmc.org.cn accessed on 5 June 2019 for FY-4A) were used instead of the SRF integrated over its entire wavelength range, of which the calculation process was time-consuming. Table A2 shows the input variables used for the RTM run in this study with their acronyms. The LUT was set up for five conditions similar to those in the study by [36] (their Figure A1). LSF1 at 0–1 km and 0–2 km without higher clouds above the LSF layer shown in Cases A and B. Cases C to E were with LSF2 accompanying the clouds. Case C and D were the clouds at 4–6 km above the 2 km fog layer, composed of either water particles or ice, respectively. Also, in Case E, clouds of ice particles exist 8–10 km above LSF1. We obtained simulated values from the LUT based on the input data of three sun or satellite orbit angles (SZA, VZA, and RAA) for each LSF case.

### 2.4. Formulation of Probability Index (PI)

Figure 3 shows the flow diagram for LSF detection near Japan at dawn during the period from March to November of 2018–2019, based on the near-simultaneous satellite observations of Himawari-8 and FY-4A. After collocating 31 ground station sites onto satellite observation grids, satellite data were collected when fog occurred at the ground stations. The optimal thresholds of eight satellite-derived variables were derived from iteration processes to discriminate between LSF and clear-sky cases based on a skill-score test. The variables were $BTD_{13.5-8.5}$, $\Delta NDSI$, $R_{0.65}$, $\Delta BTD_{3.7-11}$, $LST-BT_{11}$, $\Delta R_{0.65}$, $BTD_{3.7-11}$, and NDSI. Using the top four skill-score variables, a total of 16 cases could be extracted from their statistical combinations (Tables 3 and 4). In addition, the weights were given and merged into a total of five classes for the probability of LSF occurrence for the 16 cases. To calculate PI in real time, calculation of five PI classes and the AP (i.e., assigned probability) value in each class were conducted using the long-term data of the other training period (i.e., the 2018 control period in this current study). The introduction of the AP values is necessary to quantify the LSF occurrence by its probability. The magnitude order of AP was set based on the top four "POD minus FAR" values in the skill score test of the pre-processing. "POD minus FAR" could comprehensively explain both POD and FAR. This "PI formulation" is explained in detail in Section 3.3.

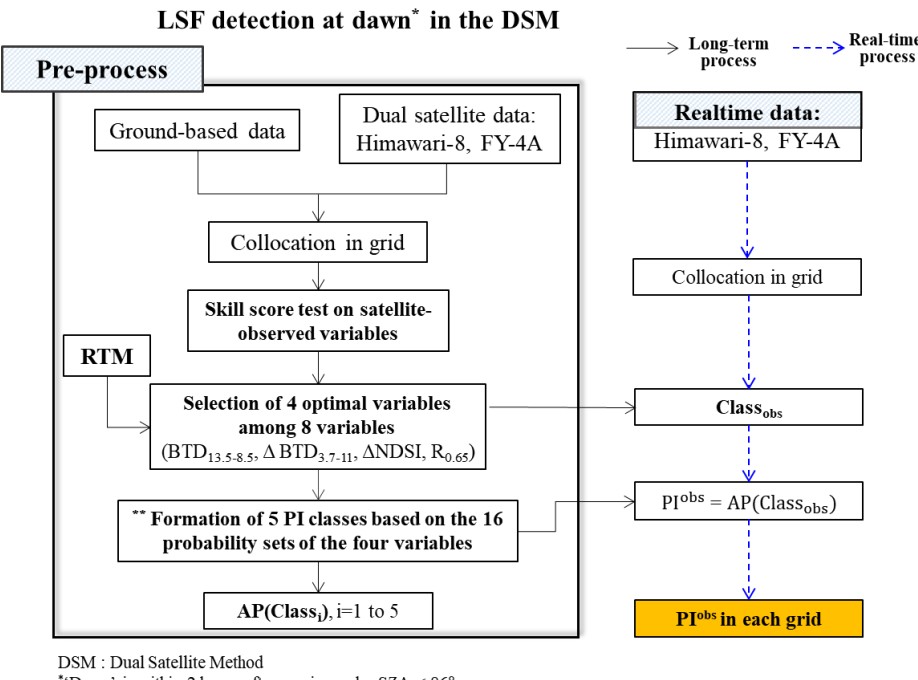

**Figure 3.** Flow diagram for LSF detection in terms of probability index (PI) near Japan at dawn during the period of March to November of 2018–2019, based on near-simultaneous satellite observations of Himawari-8 and FY-4A. The meaning of the acronyms in the diagram is explained in Table A1.

**Table 3.** Statistical verification of eight threshold values for LSF detection at dawn near Japan from the contingency table during the period of March to November of 2018, using the Himawari-8 and FY-4A satellites. POD: Probability of Detection, CSI: Critical Success Index, HSS: Heidke Skill Scores, PC: Percentage Correct, FAR: False Alarm Ratio.

| Satellite-Derived Threshold | Skill Score | | | | | |
|---|---|---|---|---|---|---|
| | POD | CSI | HSS | PC | FAR | POD-FAR |
| $-24\,\text{K} < \text{BTD}_{13.5\text{-}8.5} < -10\,\text{K}$ | 0.881 | 0.833 | 0.785 | 0.895 | 0.061 | 0.820 |
| $-0.1 < \Delta\text{NDSI} < 0.3$ | 0.881 | 0.829 | 0.778 | 0.892 | 0.066 | 0.815 |
| $0.19 < \text{R}_{0.65} < 0.52$ | 0.847 | 0.819 | 0.774 | 0.888 | 0.039 | 0.808 |
| $7\,\text{K} < \Delta\text{BTD}_{3.7\text{-}11} < 19\,\text{K}$ | 0.847 | 0.805 | 0.753 | 0.878 | 0.057 | 0.790 |
| $4.5\,\text{K} < \text{LST-BT}_{11} < 37.5\,\text{K}$ | 0.699 | 0.575 | 0.372 | 0.692 | 0.236 | 0.463 |
| $0.29 < \Delta\text{R}_{0.65} < 0.55$ | 0.460 | 0.436 | 0.337 | 0.644 | 0.110 | 0.350 |
| $3\,\text{K} < \text{BTD}_{3.7\text{-}11} < 15\,\text{K}$ | 0.466 | 0.391 | 0.167 | 0.566 | 0.293 | 0.173 |
| $-0.18 < \text{NDSI} < 0.20$ | 0.330 | 0.309 | 0.200 | 0.559 | 0.171 | 0.159 |

Utilizing these climatological values during the training period, a threshold test of the real-time dual satellite observations was carried out and provided the PI class (or no fog) in a spatial grid. The AP as a function of PI class (i.e., Class$_{obs}$) was equivalent to the PI$_{obs}$ value, indicating the LSF probability from the real-time dual satellite observations. Fog detection methods from simultaneous observations of multiple satellites have been developed in two previous studies [9,36]. We analyzed various channel information obtained from advanced satellites and tried to present the probability index of fog occurrence more precisely in this study compared to previous studies. Table 5 shows a summary of the DSM differences between previous and the present studies.

**Table 4.** Five LSF classes derived from 16 cases of the four variables (A: $BTD_{13.5-8.5}$, B: $\Delta NDSI$, C: $R_{0.65}$, D: $\Delta BTD_{3.7-11}$) based on the threshold test for detection of either LSF or clear-sky phenomena. For instance, Case 1, which passed the test for LSF detection by all four variables, corresponds to the "very high" probability of Class 1 in terms of the probability set "$P(A \cap B \cap C \cap D)$". The superscript "c" in $A^C$, which consists of the elements that are not in A in set theory, means the complement of a set A.

| Assigned Probability for 16 Cases | LSF Class (Possibility of Occurrence) | | | | |
|---|---|---|---|---|---|
| | 1 (Very High) | 2 (High) | 3 (Medium) | 4 (Low) | 5 (None) |
| 1 | $P(A \cap B \cap C \cap D)$ | | | | |
| 0.75 | | $P\left(A \cap B \cap C \cap D^C\right)$ | | | |
| 0.75 | | $P\left(A \cap B \cap D \cap C^C\right)$ | | | |
| 0.75 | | $P\left(A \cap C \cap D \cap B^C\right)$ | | | |
| 0.75 | | $P\left(B \cap C \cap D \cap A^C\right)$ | | | |
| 0.5 | | | $P\left(A \cap B \cap C^C \cap D^C\right)$ | | |
| 0.5 | | | $P\left(A \cap C \cap B^C \cap D^C\right)$ | | |
| 0.5 | | | $P\left(A \cap D \cap B^C \cap C^C\right)$ | | |
| 0.5 | | | $P\left(B \cap C \cap A^C \cap D^C\right)$ | | |
| 0.5 | | | $P\left(B \cap D \cap A^C \cap C^C\right)$ | | |
| 0.5 | | | $P\left(C \cap D \cap A^C \cap B^C\right)$ | | |
| 0.25 | | | | $P\left(A \cap B^C \cap C^C \cap D^C\right)$ | |
| 0.25 | | | | $P\left(B \cap A^C \cap C^C \cap D^C\right)$ | |
| 0.25 | | | | $P\left(C \cap A^C \cap B^C \cap D^C\right)$ | |
| 0.25 | | | | $P\left(D \cap A^C \cap B^C \cap C^C\right)$ | |
| 0 | | | | | $P\left(A^C \cap B^C \cap C^C \cap D^C\right)$ |

**Table 5.** Comparison of the results of this study with those of two previous studies of the Dual Satellite Method (DSM) for LSF detection at dawn [9,36].

| Details | DSM Study | | |
|---|---|---|---|
| | Yoo et al. (2018) [9] | Yang et al. (2019) [36] | This Study |
| Geostationary satellites (Number of channels) | COMS (6) and FY-2D (6) | COMS (6) and FY-2D (6) | Himawari-8 (16) and FY-4A (14) |
| Main research area | South Korea | South Korea | Japan |
| Spatial grid resolution | ~4 km | ~4 km | ~4 km |
| Satellite viewing zenith angle (VZA) difference at nadir | $46.5°$ | $46.5°$ | $40.4°$ |
| Period | 2013–2016 | 2013–2016 | 2018–2019 |
| Season | June to August | April to August | March to November |
| DSM variable | $\Delta R_{0.67}$ | $\Delta R_{0.67}$ and $\Delta BTD_{3.7-11}$ | $\Delta BTD_{3.7-11}$ and $\Delta NDSI$ |
| Additional variable | Suggests $R_{0.67}$ | $R_{0.67}$ | $BTD_{13.5-8.5}$ and $R_{0.65}$ |
| Probability index of LSF | No | Yes | Yes |
| Number of variables for PI calculation | None | Three | Four |
| Number of LSF occurrence classes | 3 or 4 | 8 | 5 extensible to 16 * |

\* The maximum class will be available based on the probability combination of the four variables (Table 4) if the long-term dataset of the LSF and clear-sky occurrences is given in the future.

The maximum number of PI class (i.e., 16) will be available based on the probability of the four variables if the long-term dataset of LSF and clear-sky occurrences is given in the future (Table 4).

## 3. Results

The processes for deriving the optimal thresholds and their verification will be discussed in Sections 3.1 and 3.2, based on two-year observations and the RTM simulation. These sections explain the selection of optimal parameters for fog detection. The process of developing PI and its application to case studies to address the PI merits in LSF detection are presented in Section 3.3.

### 3.1. OptimalLSF Thresholds from the 2018 Control Data

To derive optimal LSF thresholds, the average values with standard deviation ($\pm 1\sigma$) of satellite observations were investigated for LSF and clear-sky cases at dawn during the control (or training) period of 2018 (Figure 4). The seven satellite-observed values ($BTD_{13.5-8.5}$, $R_{0.65}$, $\Delta NDSI$, $\Delta BTD_{3.7-11}$, NDSI, $\Delta R_{0.65}$, and $BTD_{3.7-11}$) and the values of LST-$BT_{11}$ with ground observations of land surface temperature were analyzed to determine the range of each variable in LSF and clear-sky conditions. With the separation of LSF from the clear sky, the LSF threshold can be relatively reliable for LSF detection. Separation without overlap of the weather phenomena was noted for $R_{0.65}$, which was traditionally utilized for fog detection. In addition to the two variables ($\Delta NDSI$ and $\Delta BTD_{3.7-11}$) derived from the dual-satellites of Himawari-8 and FY-4A, LSF and clear sky were distinct in the $BTD_{13.5-8.5}$ available from Himawari-8 only. This result suggests that the threshold values derived from dual satellites ($\Delta NDSI$ and $\Delta BTD_{3.7-11}$) can be worthy for LSF detection in addition to single satellite observations ($BTD_{13.5-8.5}$ and $R_{0.65}$). However, other thresholds (LST-$BT_{11}$, $BTD_{3.7-11}$, $\Delta R_{0.65}$, and NDSI) have uncertainties for LSF detection due to mixed information that cannot discriminate between LSF and clear sky.

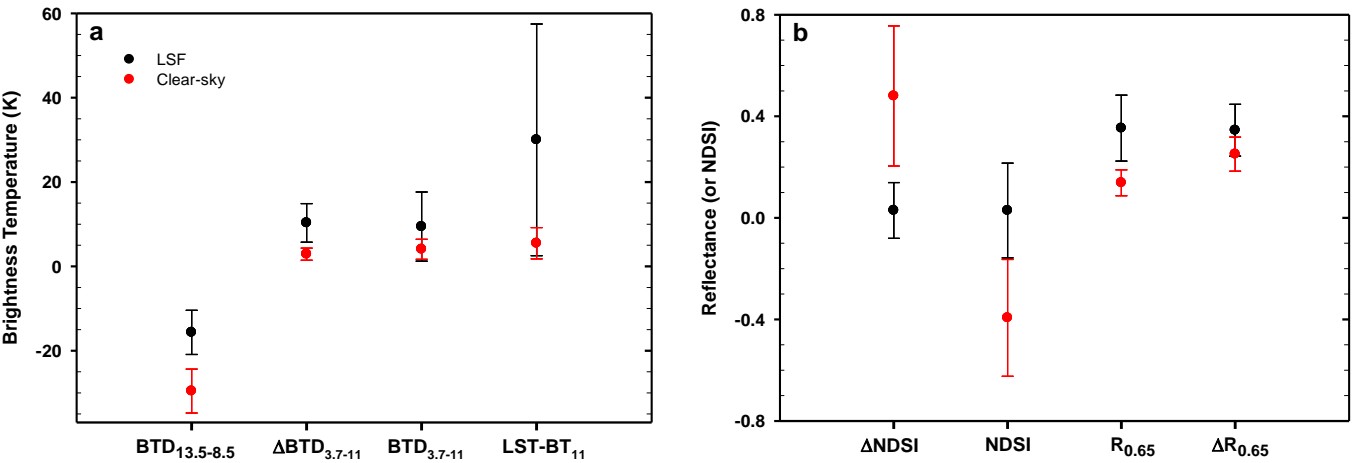

**Figure 4.** Average values of eight satellite-observed variables with their standard deviation ($\pm 1\sigma$) for two weather conditions (black for LSF and red for clear sky) at dawn during the control period of 2018 near Japan. (**a**) Brightness temperatures ($BTD_{13.5-8.5}$, $\Delta BTD_{3.7-11}$, $BTD_{3.7-11}$, and LST-$BT_{11}$). (**b**) Reflectances (or NDSI) of $\Delta NDSI$, NDSI, $R_{0.65}$, and $\Delta R_{0.65}$.

Using the data for the control period of 2018, the optimal thresholds were derived through the iteration process. The lower boundary value (i.e., $Th_{lower}$ in Table 6) that distinguished LSF and clear sky was obtained through iteration, while the upper boundary value (i.e., $Th_{upper}$) was adjusted to the +1 $\sigma$ range of the average value to avoid middle or high cloud contamination. The results of statistical verification for each threshold variable are shown in the contingency table and the index definition (Table A3), based on the five skill score indices (HSS, CSI, POD, PC, and FAR) shown in Table 3. HSS can be an overall skill indicator of detection accuracy [52], although the difference between POD and FAR (i.e., POD minus FAR) was analyzed in this study as an important indicator of detection accuracy. The difference values were higher in magnitude order for $BTD_{13.5-8.5}$, $\Delta NDSI$, $R_{0.65}$, and $\Delta BTD_{3.7-11}$. In these four variables, the skill scores of POD minus FAR (0.79–0.82) were roughly the same, within a 3% difference. Meanwhile, the POD minus FAR values for

the other four variables did not exceed 0.47. Therefore, the threshold values of the top four variables were more useful for LSF detection than were those of the other four thresholds.

**Table 6.** Satellite-derived PI distributions of five classes weighted at 0.25 intervals for detection of LSF and clear-sky cases in 2019.

| LSF Class | Assigned Probability | Th$_{lower}$ < LSF < Th$_{upper}$ | | Either Clear Sky < Th$_{lower}$ or Clear Sky > Th$_{upper}$ | |
|---|---|---|---|---|---|
| | | Number of Actual Occurrences | Sum of Detection Probability | Number of Actual Occurrences | Sum of Detection Probability |
| 1 | 1 | 99 | 99.00 | 79 | 79.00 |
| 2 | 0.75 | 43 | 32.25 | 8 | 6.00 |
| 3 | 0.5 | 21 | 10.50 | 12 | 6.00 |
| 4 | 0.25 | 2 | 0.50 | 11 | 2.75 |
| 5 | 0 | 6 | 0.00 | 0 | 0.00 |
| | Total | 171 | 140.25 | 110 | 93.75 |

Figure 5 shows scatter plots of the top four variables (BTD$_{13.5-8.5}$, $\Delta$BTD$_{3.7-11}$, $\Delta$NDSI, and R$_{0.65}$) with respect to LST-BT$_{11}$ for the weather phenomena of LSF (red circle) and clear sky (blue triangle) near Japan at dawn during 2018. The value of LST-BT$_{11}$ tended to be higher under middle/high clouds than in LSF/clear-sky conditions. The weather phenomena of LSF and clear sky were categorized well using the top four variables, with high verification scores of POD minus FAR.

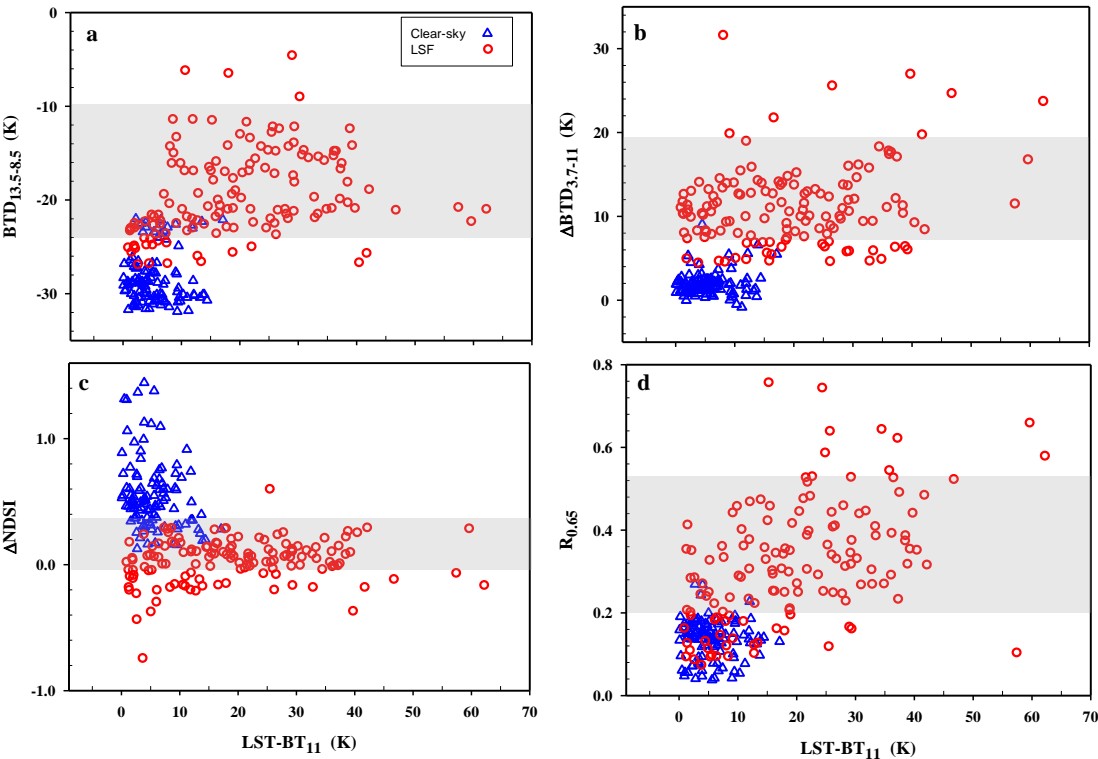

**Figure 5.** Scatter diagrams of (**a**) BTD$_{13.5-8.5}$, (**b**) $\Delta$BTD$_{3.7-11}$, (**c**) $\Delta$NDSI, and (**d**) R$_{0.65}$ with respect to LST-BT$_{11}$ for the weather phenomena of LSF (red circle) and clear sky (blue triangle) at dawn during 2018 near Japan. The values in the ordinate of the gray-shaded bands indicate the LSF threshold ranges for each variable. The satellite-observed value of one positive standard deviation (i.e., +1$\sigma$; upper boundary) for LSF detection is used to remove the middle or high-level clouds without accompanying LSF, except for $\Delta$NDSI. The negative value of -1$\sigma$ (lower boundary) is needed for $\Delta$NDSI of Figure 5c based on the results shown in Figure 4b.

To theoretically support these observational results, RTM simulation was performed considering various optical characteristics (e.g., altitude and optical thickness of the fog layer, altitude and optical thickness of upper clouds, effective radii of fog/cloud particles) in Table A2, and the results were shown in Figure 6. The figure shows the simulation results in the domains of a) $BTD_{13.5-8.5}$ vs. $LST-BT_{11}$, b) $\Delta BTD_{3.7-11}$ vs. $LST-BT_{11}$, c) $\Delta NDSI$ vs. $LST-BT_{11}$, and d) $R_{0.65}$ vs. $LST-BT_{11}$. The average values of the Himawari-8 angles (i.e., SZA = 79.5°, RAA = 74.5°, and VZA = 41.5°) for 176 LSF cases during 2018 were used as RTM input. The satellite-observed mean values are included in the figure for LSF (black asterisk) and clear sky (gray-cross). The clear-sky values from the observations and simulations (pink asterisk) are depicted using black circles to show the approximate agreement between them. Five colors (yellow, blue, red, navy, and green) denote the heights (FH or CH: 1, 2, 4–6, 8–10 km) of the fog and upper cloud layers with respect to effective radii (ER: 2, 4, 8, 16 μm) and water/ice phase. The RTM details are given in Table A2.

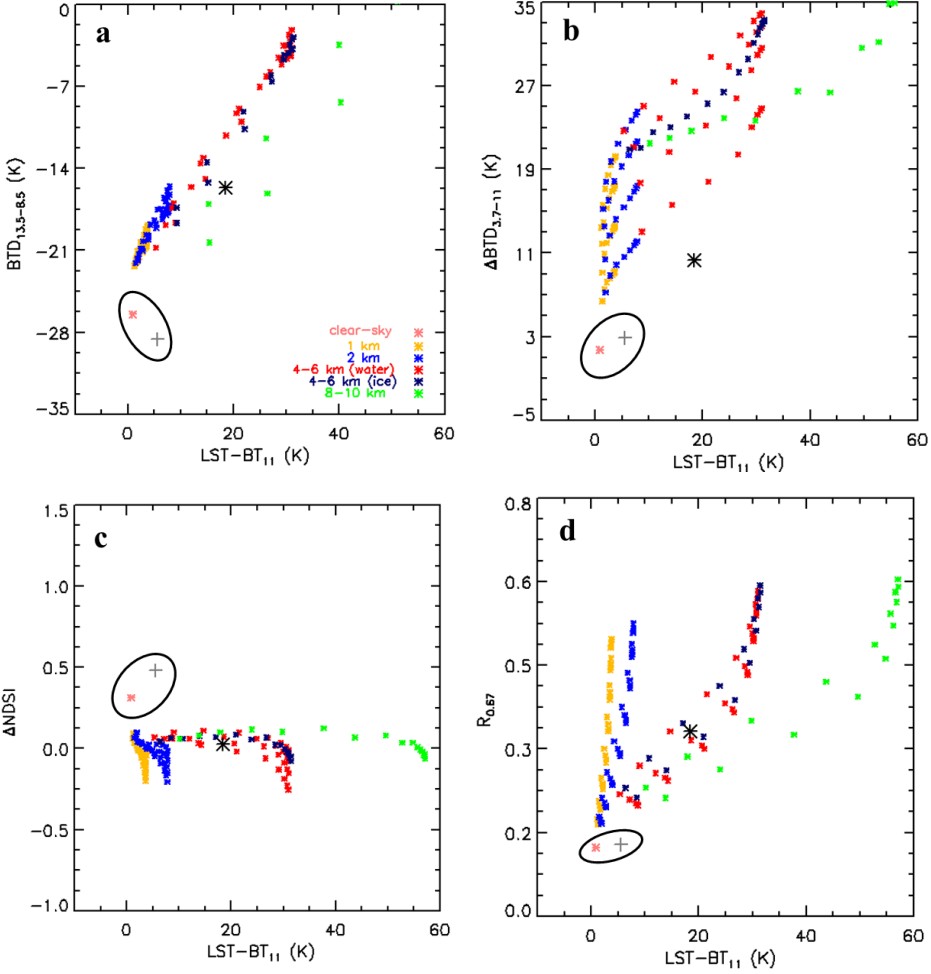

**Figure 6.** RTM LSF simulation in the domain of (**a**) $BTD_{13.5-8.5}$ vs. $LST-BT_{11}$, (**b**) $\Delta BTD_{3.7-11}$ vs. $LST-BT_{11}$, (**c**) $\Delta NDSI$ vs. $LST-BT_{11}$, and (**d**) $R_{0.67}$ vs. $LST-BT_{11}$ at dawn during 2018 near Japan under the different LSF conditions. The yellow and blue symbols in the figures denote the fog layer at 0–1 km and 0–2 km without higher clouds. Middle clouds of water/ice at 4–6 km height are shown in red and navy, respectively. Green asterisk means higher cloud (8–10 km) above the 0–2 km fog layer. The average values of the Himawari-8 angles (i.e., SZA, RAA, and VZA) for 176 LSF cases were used as RTM input. The RTM conditions are described in Table A2. The satellite-observed mean values are shown for LSF (black asterisk) and clear sky (gray-cross). The clear-sky values from the observations and simulations (pink asterisk) are depicted in black circles. Some simulated data out of range in the domain were not shown for a comparison with the observations of Figure 5.



The DSM variables (i.e., $\Delta$NDSI and $\Delta$BTD$_{3.7-11}$), as well as BTD$_{13.5-8.5}$ and R$_{0.65}$ from the single satellite observations (i.e., Himawari-8 only), had a significant tendency to separate LSF and clear sky at dawn (Figure 6a–d) due to the unique satellite orbit in space (i.e., VZA difference between the two satellites). The distributions of observed mean values for the two meteorological phenomena (LSF and clear sky) were consistent within the ordinate range of the simulation. The difference in "LST-BT$_{11}$" between observations (5.49 K) and simulations (0.82 K) for clear-sky conditions could be explained in terms of the effect of water vapor and optically thin clouds during the warm season [53]. Considering that the LST-BT$_{11}$ value is 19.7 K for LSF and 5.5 K for clear sky, the average LSF top height is 2–2.5 km, consistent with that of [11].

However, the simulated values of $\Delta$BTD$_{3.7-11}$ on the abscissa were somewhat out of range with the average of its observations (Figure 6b). According to the simulation, the upper clouds above the fog layer could cause errors in the LSF detection of satellite observations. The upper clouds generally show relatively large values of LST-BT$_{11}$, compared to the lower clouds. Overall, the simulations were in good agreement with fog detection in DSM.

## 3.2. Verification of Satellite-Observed Thresholds during the Experimental Period of 2019

The eight LSF threshold values derived from the control period of 2018 for LSF detection at dawn were applied to the experimental data during 2019. Figure 7 shows statistical verification for the threshold values of eight satellite-observed variables in terms of a) POD minus FAR, b) POD, c) FAR, and d) CSI. The verification was performed based on the total ground-observations of 171 fog and 110 clear-sky occurrences at dawn near Japan. The thresholds of the top four variables (BTD$_{13.5-8.5}$, R$_{0.65}$, $\Delta$NDSI, and $\Delta$BTD$_{3.7-11}$) in the verification skill-scores were substantially better than those of the other four variables. The "POD minus FAR" scores of the top four variables during the experimental period were lower by ~5% than those during the training period (Table 3). In particular, the "$\Delta$NDSI" threshold was most sensitive to the interannual dependence of threshold accuracy. However, more long-term data seemed to be required to estimate interannual variability in view of the two years of data in this study.

Figure 8 presents the "$\pm1\sigma$" range of LSF and clear-sky cases for each variable in the same way as Figure 4 so that the biennial changes in satellite-observed variables under the weather phenomena were obtained from a comparison of the results shown in the two Figures. Since the threshold values were optimized for the training period data, statistical verification results decreased slightly (Figure 7 and Table 3). However, there were no significant differences in skill scores for the top four variables (BTD$_{13.5-8.5}$, $\Delta$BTD$_{3.7-11}$, $\Delta$NDSI, and R$_{0.65}$) between the two years. In addition, the interannual change of NDSI among the entire eight variables was noticeable. Although the skill scores of NDSI itself were not good, $\Delta$NDSI showed excellent results due to the DSM advantage. This result was due to not only the large VZA difference between the two satellites but also to the relatively long optical path of FY-4A, which was viewed from outside of the nadir. Therefore, the DSM was likely to cancel the noise error and amplify the signal for LSF detection. This tendency was similar to that of the $\Delta$BTD$_{3.7-11}$ case in a previous study [36].

## 3.3. PI development and Case Study

The probability index of LSF was developed to present fog occurrence as a probability concept rather than an alternative one (either no fog or fog), using the top four variables that showed excellent results (POD ~0.8; POD minus FAR ~0.7) during the experimental period. In Table 4, five LSF classes were derived from 16 probability sets of the variables based on the threshold test for detection of either LSF or clear-sky phenomenon. For instance, Case 1, which passed the test for LSF detection by all four variables, corresponded to Class 1 in terms of the set "P(A $\cap$ B $\cap$ C $\cap$ D)" with an assigned probability of 1.0. For Class 1, in which all four variables passed their own thresholds, the probability of LSF occurrence was classified as "very high". In the same way, three threshold values among

the four variables were achieved for those in Class 2. In addition, any combination of the three variables was classified as a "high" possibility of LSF occurrence. In this regard, the number of cases that passed the threshold test for the four satellite-observed variables (i.e., $BTD_{13.5-8.5}$ in gray rectangle, $\Delta NDSI$ in orange, $\Delta BTD_{3.7-11}$ in blue, and $R_{0.65}$ in green) was shown in the detection of a) LSF and b) clear sky in the so-called Venn diagram shown in Figure 9. In other words, the zones of red (Class 1 for the occurrence possibility of the two weather phenomena), yellow (Class 2), blue (Class 3), and white (Class 4) were defined and were also explained in the Table.

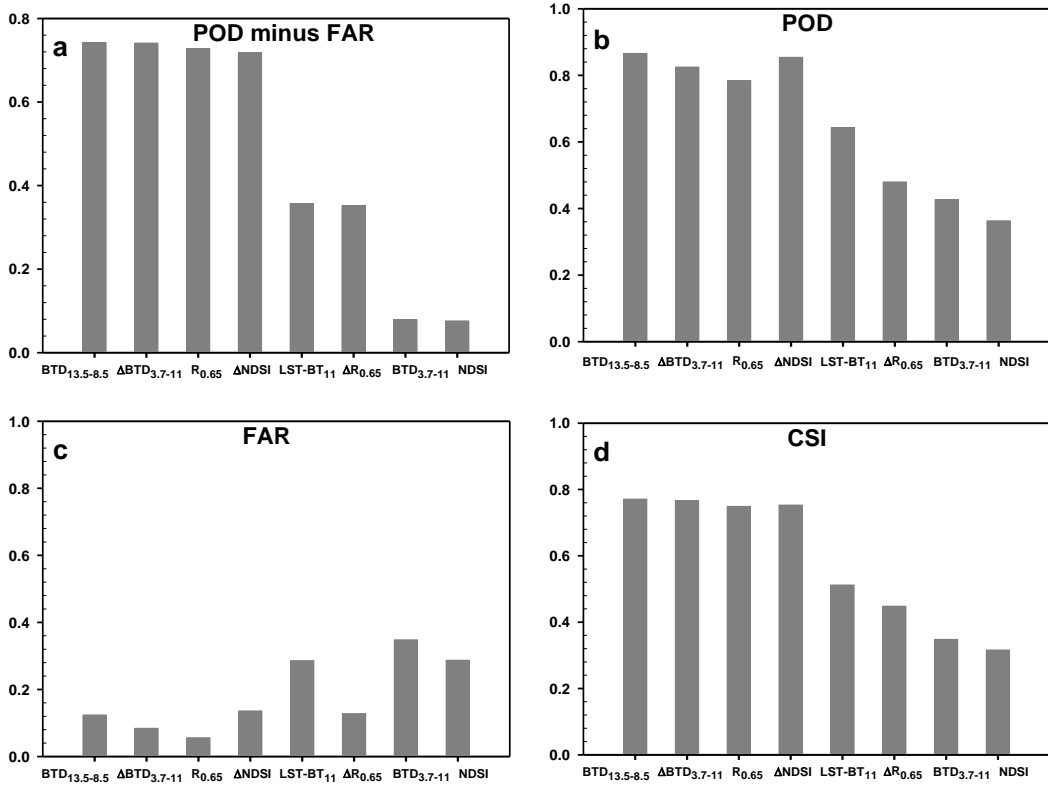

**Figure 7.** Statistical verification of (**a**) POD minus FAR, (**b**) POD, (**c**) FAR, and (**d**) CSI with respect to the LSF threshold values of eight satellite-observed variables during the experimental period of 2019. The verification was performed based on the total ground-observations of 171 fog and 110 clear-sky occurrences at dawn near Japan.

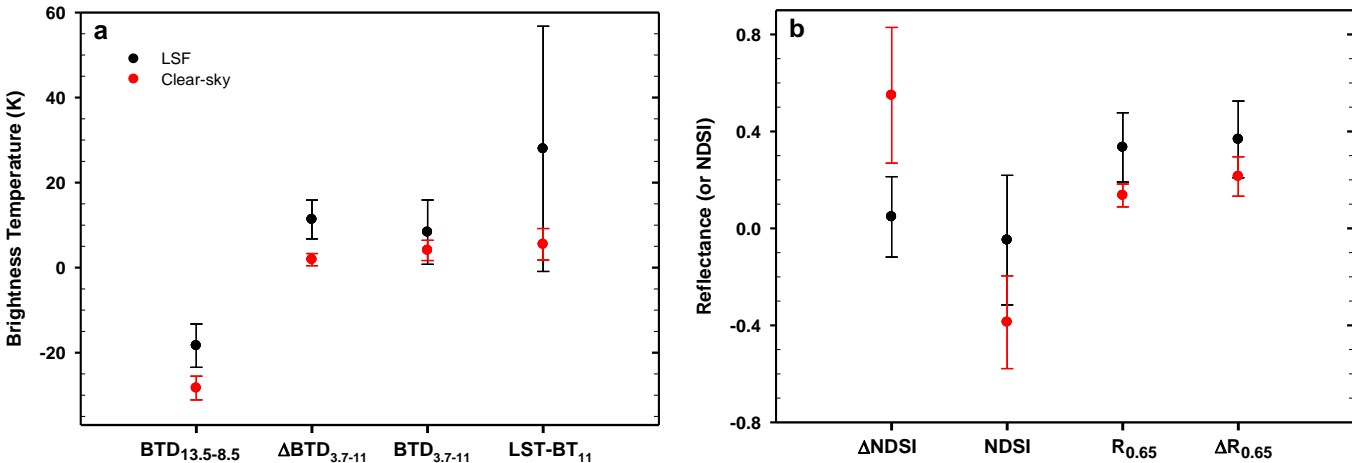

**Figure 8.** (**a**) Same as Figure 4a except for the period of 2019. (**b**) Same as Figure 4b except for the period of 2019.

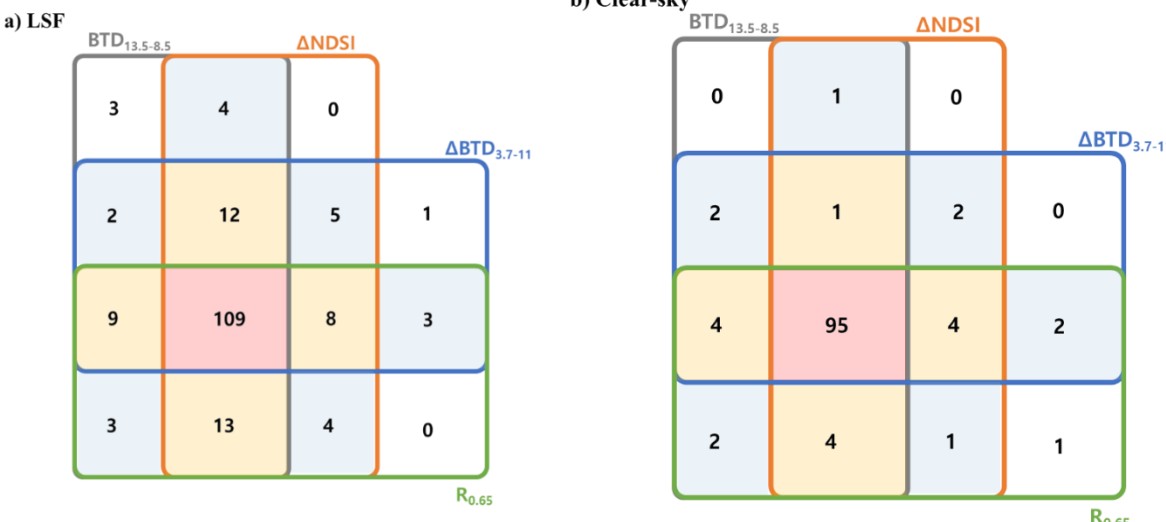

**Figure 9.** The number of cases that have passed the threshold test of the four satellite-observed variables (i.e., $BTD_{13.5-8.5}$ in gray rectangle, $\Delta NDSI$ in orange, $\Delta BTD_{3.7-11}$ in blue, and $R_{0.65}$ in green) for detection of (**a**) LSF and (**b**) clear sky at dawn during 2019 near Japan. The zones of red (Class 1 for the possibility of the weather phenomena), yellow (Class 2), blue (Class 3), and white (Class 4) are defined in Table 4.

When the LSF class in Table 4 was determined for a total of 171 LSF cases during the experimental period of 2019, 99 passed all four thresholds (i.e., Class 1). In addition, 43 cases passed three thresholds (i.e., Class 2). These results are described in detail in Table 6. Satellite-derived PI distributions of the five classes in the Table were weighted at 0.25 probability intervals for detection of LSF and clear-sky cases in 2019, and detection was quantitatively presented in five steps rather than two (i.e., yes or no). The probability values were also shown in terms of "assigned probability" in Table 4. If the PI results were evaluated for LSF cases during 2019, the 171 fog occurrences represented a detection score of 140.25. In clear-sky conditions, the score was 93.75 of 110 (Table 6). This indicated excellent detection of realistic PI values in the spatial distribution, rather than either "fog" or "no fog" from the traditional method. In other words, when LSF detection was performed using a single threshold, the existence of LSF was generally classified with a probability of either 0 or 1. Meanwhile, the PI in this study showed five classified probability values based on the four satellite-observed variables in DSM, for useful and reliable LSF detection.

The PI in the case study was applied to two foggy scenes at dawn near Japan (128–145 E, 30–45 N) to compare the results with those of the conventional method (i.e., using a single threshold value of either $R_{0.65}$ or $BTD_{3.7-11}$) in the LSF spatial distribution. Figure 10 shows the fog probability distribution at dawn (06:00 LST) on July 9, 2019, near Japan using the detection methods of a) $R_{0.65}$ threshold, b) $BTD_{3.7-11}$ threshold, and c) LSF PI in this study. The $BT_{11}$ distribution for approximate cloud-top temperature and the SYNOP map for station-observed fog occurrences were shown in Figure 10d,e, respectively. Fog occurrences (pink triangles in Figure 10a–d) were reported (station numbers 20, 26, and 31 in Table 2) at three stations by ASOS (automated surface observing system).

The $R_{0.65}$ threshold successfully detected two of the three fog occurrences at a detection rate of 0.66 (Figure 10a); the rate was 0.33 with the $BTD_{3.7-11}$ threshold (Figure 10b). The PI produced by combining the four variables using dual-satellite observations showed a probability of 0.66, higher than that in the $BTD_{3.7-11}$ case but similar to that in the $R_{0.65}$ case. Although PI accuracy was the same as that of the $R_{0.65}$, it provided LSF spatial distribution in terms of probability in five classes. To examine the LSF-related information over the study area, the $BT_{11}$ distribution and the weather map chart were analyzed simultaneously (Figure 10d,e). The $BT_{11}$ was useful to estimate the approximate cloud top temperature. In addition, the map allowed the identification of the distributions of foggy and humid areas

and lower or upper clouds. The $BT_{11}$ values were either high ($\geq$ ~290 K) or low ($\leq$ ~230 K) in areas where PI indicated no or low probability of fog due to almost clear-sky or optically thin cloud conditions. In Figure 10c, the LSF PI values indicated that the LSF occurrences were "very high" or "high" in areas where fog actually occurred on the weather map (pink triangle). For areas at station numbers 20 and 31 where fog detection failed using $R_{0.65}$ or $BTD_{3.7\text{-}11}$, PI indicated a relatively low probability of fog but was the only method to generally succeed in the detection (PI = 0.75).

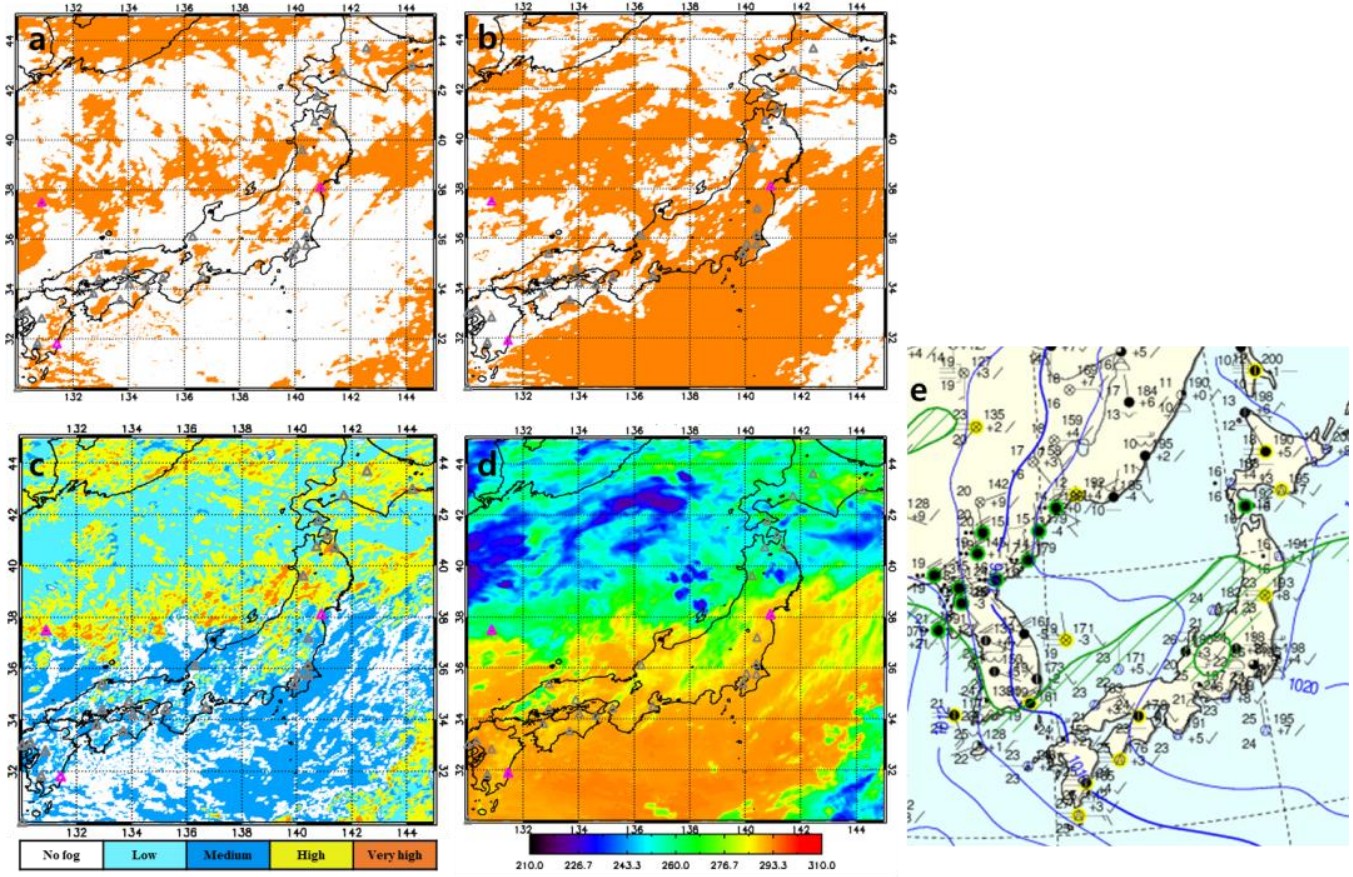

**Figure 10.** Spatial distributions of fog probability at dawn (06:00 LST) on 9 July 2019, near Japan from detection methods of (**a**) $R_{0.65}$ threshold, (**b**) $BTD_{3.7\text{-}11}$ threshold, and (**c**) LSF PI in this study. (**d**) $BT_{11}$ distribution for cloud-top temperatures, and (**e**) SYNOP map for station-observed fog occurrences. Fog occurrences at the stations (pink triangles in Figure 9a,b) were reported by ASOS.

Figure 11 is the same as Figure 10 except for 05:30 LST on 4 July 2019, and without a SYNOP map (not available at the time). According to the ASOS report, there were three fog occurrences at the stations (in Table 2; 6, 16, and 27). The time difference between the fog occurrence on the report and that on the weather map was greater than 30 minutes, so the low cloud and fog areas were analyzed with the distribution of $BT_{11}$ without the map. The PI and $R_{0.65}$ successfully detected all three fog occurrences, showing a detection probability of one. However, the $BTD_{3.7\text{-}11}$ threshold detected only one of the three, a probability of 0.33, only successful at station 6. These results show that PI was able to detect fog more accurately than $BTD_{3.7\text{-}11}$ and was more realistic (or natural) and excellent than $R_{0.65}$ in the LSF spatial distribution. In particular, detailed information on the LSF PI in five classes was useful for aviation and navigation. So far, it has been difficult to identify the occurrence of sea fog in the open ocean despite its frequent occurrence due to the limited number of ground observations. The spatial PI distribution of LSF over a large area could be useful to monitor fog phenomena over an extensive region including the ocean in almost real-time.

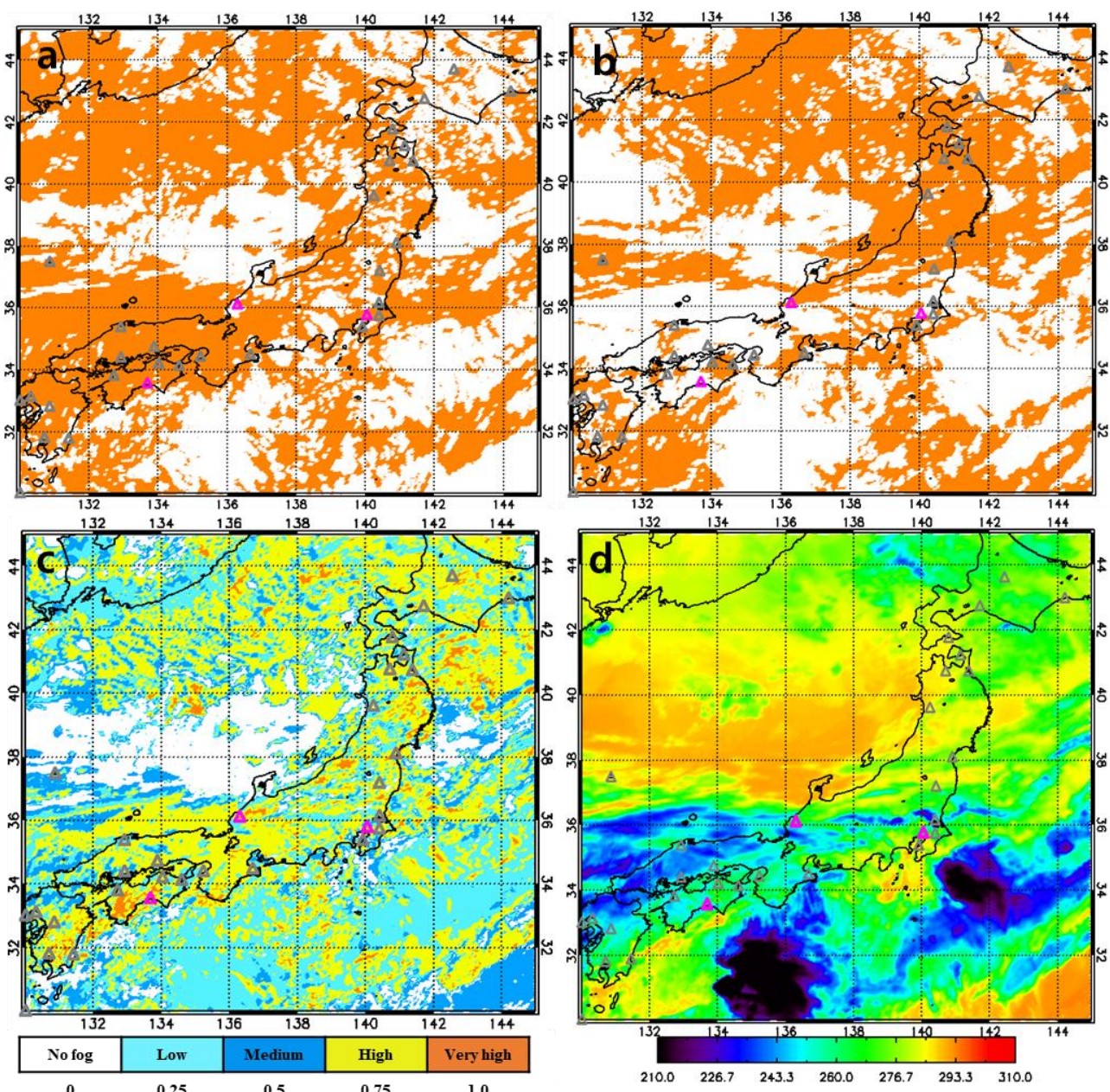

**Figure 11.** (**a**) Same as Figure 10a except for the date and time (05:30 LST on 4 July 2019), (**b**) Same as Figure 10b except for the date and time. (**c**) Same as Figure 10c except for the date and time. (**d**) Same as Figure 10d except for the date and time, and no SYNOP map (not available at the time).

## 4. Discussion

The DSM used in this study is applicable to LSF (or fog) detection based on advanced satellite observations, including GEO-KOMPSAT-2A (GK2A; [54]) in the future, for real-time monitoring. Further improvement in nowcasting and forecasting (e.g., advective fog) can be expected. The Advanced Meteorological Imager (AMI) data of GK2A will be useful for the LSF analysis over the Korean Peninsula because of its nadir point. The DSM from FY-4A and Himawari-8 were applied to the LSF detection near Japan at dawn due to the Himawari-8 nadir. Similarly, the method can be utilized for the detection over the inland region of China under the FY-4A nadir at dusk, although LSF generally occurs at dusk less than at dawn due to daytime solar heating.

Various applications of the DSM in the meteorological and environmental fields are possible to observe fog targets in a three-dimensional structure from a stereoscopic view,

similar to observations with two eyes [37]. Thus, this method can be applied to any worldwide region where simultaneous observations by two geostationary-orbit satellites are available without additional expense. However, the difference in VZA between the two satellites is required to be between $40°$ and $50°$ to achieve optimal LSF detection. For this reason, the detection accuracy of $\Delta R_{0.65}$, which had been excellent in the COMS and FY-2D (VZA difference ~46.5°) [9], was significantly reduced in this study. It seems that $\Delta R_{0.65}$ is more sensitive to both VZA and seasonality than to the top-four variables in this study. Here the variables of $\Delta NDSI$ and $\Delta BTD_{3.7-11}$ were newly utilized for LSF detection in terms of PI, despite the decrease in VZA, with the $BTD_{13.5-8.5}$ from the spectral difference of Himawari-8 only.

The RTM simulation of this study may have some limitations in the diverse weather conditions of multilayer, broken, or mixed-phase clouds [9]. Also, errors in the LSF detection due to the higher (or upper) clouds above the LSF layer have been discussed in detail, based on the simulation [9,36]. However, the adverse issues may not be so critical for the LSF (under ~2.5km height) in relatively stable atmospheric conditions, considering the ratio (13–20%) of satellite-observed multi-layer cloudy pixels to all cloudy pixels [55] and the vertical properties of ice-over-water clouds [56]. In addition, for a comparison of the two GEO data in this study, the coarse-grid averaging of Himawari-8 observations from 2 km-by-2 km to 4 km-by-4 km may miss the detection of optically thin LSF phenomena in a small horizontal scale less than 4 km, particularly in the tests of $BTD_{13.5-8.5}$ and $R_{0.65}$.

## 5. Conclusions

In this study, the dual-satellite method was applied to advanced satellites (Himawari-8 and FY-4A), and newly developed thresholds with additional channels and LSF spatial distribution in terms of PI were provided to improve its detection in Japan at dawn. In addition, the optical characteristics of LSF were estimated using the LUT from the RTM simulation under various conditions and were consistent with satellite-derived observations. The PI of LSF in the DSM was derived by combining the major threshold components (i.e., $BTD_{13.5-8.5}$, $\Delta NDSI$, $\Delta BTD_{3.7-11}$, and $R_{0.65}$) based on the top-four "POD minus FAR" scores ($\geq 79\%$) among the eight variables.

The case study was carried out to address the PI merits of the LSF spatial distribution. The PI values in this study were similar to $R_{0.65}$ simply based on the success rate of fog detection due to the use of $R_{0.65}$ as one of the four equally-weighted variables for constructing the PI. Thus, the fog detection rate calculated from PI was expected to be similar to that from $R_{0.65}$. However, the spatial distributions revealed the advantages of PI. Since multi-stage fog detection was possible in PI over a wide area, the LSF distribution in the DSM could be presented in more detail than that of the conventional methods of $R_{0.65}$ and/or $BTD_{3.7-11}$. More specifically, $R_{0.65}$ only reported two classes of either presence or absence of fog, whereas PI presented fog occurrence probability as five classes. The classes can be increased to a maximum of 16 if a long-term dataset is available in the future, as shown in the statistical combination (Table 4). In this case, different probability weights would be assigned with respect to the variables. Thus, the LSF PI can be used as more practical and useful information for navigation and aviation. In particular, we believe that LSF PI is valuable for open oceanic areas where ground observations are limited in view of both accuracy and spatial distribution.

**Author Contributions:** Conceptualization, J.-H.Y., J.-M.Y.; methodology, J.-H.Y., J.-M.Y.; software, J.-H.Y.; validation, J.-H.Y. and J.-M.Y.; formal analysis, J.-H.Y. and J.-M.Y.; investigation, J.-H.Y. and J.-M.Y.; data curation, J.-H.Y.; writing—original draft preparation, J.-H.Y. and J.-M.Y.; writing—review and editing, J.-H.Y., J.-M.Y. and Y.-S.C.; visualization, J.-H.Y.; supervision, J.-M.Y.; project administration, J.-M.Y.; funding acquisition, J.-M.Y. All authors have read and agreed to the published version of the manuscript.

**Funding:** This work was supported by the National Research Foundation of Korea (NRF) grant funded by the Korean government (MSIT) (No. 2019R1F1A1059984).

**Acknowledgments:** We thank the employees at the CMA (for FY-4A) and the NMSC/KMA (for Himawari-8) for providing the satellite data. We are grateful to the anonymous reviewers for the constructive comments.

**Conflicts of Interest:** The authors declare no conflict of interest.

## Appendix A

**Table A1.** List of acronyms used in this study.

| Acronyms | Original Words (or Details) | Acronyms | Original Words (or Details) |
|---|---|---|---|
| AGRI | Advanced Geostationary Radiation Imager | GK2A | GEO-KOMPSAT-2A |
| AHI | Advanced Himawari Imager | GOCI | Geostationary Ocean Color Imager |
| AP | assigned probability | HR | hit rate |
| ASOS | automated surface observing system | HSS | PI estimated from satellite-observed scene |
| AVHRR | Advanced Very High-Resolution Radiometer | IR | infrared |
| AWS | automatic weather station | KMA | Korea Meteorological Administration |
| BRDF | bidirectional reflectance function | LSF | low stratus and fog |
| $BT_{11}$ | brightness temperature at ~11 μm | LUT | look-up table |
| $BT_{3.7}$ | brightness temperature at ~3.7 μm | METAR | METeorological Aerodrome Report |
| $BTD_{13.5-8.5}$ | difference between $BT_{13.5}$ and $BT_{8.5}$ | MODIS | Moderate Resolution Imaging Spectroradiometer |
| $BTD_{3.7-11}$ | difference between $BT_{3.7}$ and $BT_{11}$ | NDSI | normalized difference snow index |
| CAVOK | Ceiling (or clouds) And Visibility OK | PC | percentage correct |
| CH | cloud height | PI | probability index |
| COMS | Korean Communication, Ocean, and Meteorological Satellite | POD | probability of detection |
| COT | cloud optical thickness | $R_{0.65}$ | reflectance at ~0.65 μm |
| CSI | global telecommunications system | RAA | relative azimuth angle |
| DSM | dual satellite method | RTM | radiative transfer model |
| ER | effective radius | SIRS-B | Satellite Infrared Spectrometer |
| FAR | false alarm ratio | SRF | spectral response function |
| FG | fog | SYNOP | surface synoptic observations |
| FH | fog height | SZA | solar zenith angle |
| FOT | fog optical thickness | THLOWER | lower threshold |
| FY-2D | Chinese FengYun-2D | THUPPER | upper threshold |
| FY-4A | Chinese FengYun-4A | VIS | visible |
| GEO | geostationary-orbit satellite | VZA | satellite viewing zenith angle |

**Table A2.** Input variables of SBDART for the LUT product.

| Input variable | Contents |
|---|---|
| atmospheric profile | Mid-latitude summer, US62 |
| wavelength (λ):three channels of VIS, SWIR, & IR1 for COMS & FY-2D | 0.55–0.90, 3.5–4.0, 10.3–11.3 μm |
| solar zenith angle (SZA) | $0 \leq SZA \leq 80°$ at 10° intervals, and 85° |
| surface type | Ocean, Vegetation |
| fog height (FH) | Water fog at 0–1 km or 0–2 km |
| upper cloud height (CH) above the fog layer | Water/ice cloud (4–6 km), Ice cloud (8–10 km) |
| fog optical thickness (FOT) | 0, 0.5, 1, 2, 4, 8, 16, 32, 64 |
| cloud optical thickness (COT) | 0, 4, 8, 16, 32 |
| effective radius of fog (FER) | 4, 8, 16, 32 μm |
| effective radius of cloud (CER) | 2, 4, 8, 16 μm |
| flux computation stream | 32 |
| vertical resolution | 1 km |
| viewing zenith angle (VZA) | $0 \leq VZA \leq 90°$ at 10° intervals |
| relative azimuth angle (RAA) | $0 \leq RAA \leq 180°$ at 30° intervals |
| boundary layer aerosol type | Urban |
| **vertical optical depth of boundary layer aerosolsnominally at 0.55 μm** | **0.2** |

**Table A3.** Contingency Table and definitions for the statistical skill test.

| | | SYNOP | |
|---|---|---|---|
| | | **LSF** | **Clear sky** |
| **Satellite observation** | **LSF** | **a** | **b** |
| | **Clear sky** | **c** | **d** |

$$\text{POD} = \frac{a}{a+c} \quad \text{CSI} = \frac{a}{a+b+c} \quad \text{FAR} = \frac{b}{a+b} \quad \text{HSS} = \frac{2(ad-bc)}{(a+c)(c+d)+(a+b)(b+d)} \quad \text{PC} = \frac{a+d}{a+b+c+d}$$

$$\text{POD-FAR} = \frac{a}{a+c} - \frac{b}{a+b}$$

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
