# Peer review of "Advanced Dual-Satellite Method for Detection of Low Stratus and Fog near Japan at Dawn from FY-4A and Himawari-8"

_remotesensing, doi:10.3390/rs13051042_

Round 1
Reviewer 1 Report
The presented manuscript is devoted to dual-satellite method (DSM) for detection of the low stratus and fog (LSF) at dawn and is a logical continuation of the authors' previous research [Yoo et al Remote Sensing of Environment 2018, 87, 279; doi: 10.1016/j.rse.2018.04.019] and [Yang et al Remote Sensing 2019, 11, 1283; doi: 10.3390/rs11111283]. Both studies applied DSM to LSF detection at dawn near the Korean Peninsula. The purpose of this study was to improve the detection accuracy and spatial information of LSF at dawn near Japan in terms of probability index (PI) by devising additional tests from the advanced satellites Feng-Yun-4A and Himawari-8.
I believe that the goals and objectives of the study are clearly defined. The method for determining the corrections is described in sufficient detail. The obtained results are of undoubted interest.
To summarize, the results obtained by the authors are impressive. The method used by the authors is very promising and has a wide range of possible applications. I think that this paper could be interesting and useful for a potential reader of the "Remote Sensing" journal. The article may be accepted for publication after minor revision which takes into account the comments collected in attached file.

Author Response
Thank you for your constructive comments. We have provided a point-by-point response to the comments. Please see the attachment.

Reviewer 2 Report
The authors developed an advanced dual-satellite method for the LSF detection at dawn in terms of probability indices. Compared to traditional single satellite method, the information of different viewing angles from satellites Himawari-8 and FY-4A were taken into consideration. The new method added four more tests on the basis of the original tests, improving the detection of the LSF at dawn, and meanwhile, providing more details in LSF spatial distribution. The manuscript is well structured and clearly written, so I suggest it to be accepted after a minor revision. The following lists several concerns on this paper.
- A paper for the introduction of FY-4A is suggested to be briefly discussed (doi: 10.1175/BAMS-D-16-0065.1).
- To compare the two satellite observations, the authors collocated Himawari-8 observations to FY-4A along with the ground observations. Can you describe the collocation method in detail? As you mentioned in Lines 148-150, the analysis showed less sensitive to spatial resolution in the paper. I wonder whether the results will be influenced by the spatial inhomogeneity of LSF if you resampled Himawari-8 observations to 4km by 4km grid box by averaging?
- The authors compared the current models with two previous ones in Table 5. Is it possible to compare their performance as well. Collecting the same observations may be difficult, but the comparison may be only performed by applying the proposed method, not for the exactly same satellite. This may give us a better understanding on the importance of different tests.
- As some recent works noticed that multilayer clouds can not only be detected by radiometers (doi:10.5194/amt-13-3263-2020) but also be quantitatively retrieved (doi:10.1029/2020GL088941). Thus, it is clearly that the detection of LSF pixels may be weakly blocked by upper layer clouds as well, and how would the author think that the model performs if an upper layer cloud is existing. This is suggested to be briefly discussed.
- Line 69: Please notice that the full name of “AHI” should be given where the first abbreviation appears, and check for the entire paper (e.g. Line 80 “COMS”).
- Line 129-130: “There were two visible (VIS) channels at ~0.65 μm and ~1.6 μm.” Do you mean two solar channels? This sentence may lead to misunderstanding, please rephrase it.
Author Response

(The authors gave the same response as above.)
